# Addressing the Role of Landraces in the Sustainability of Mediterranean Agroecosystems

**Guiomar Carranza-Gallego** [1],*, **Gloria I. Guzmán** [1], **Roberto Garcia-Ruíz** [2], **Manuel González de Molina** [1] **and Eduardo Aguilera** [3]

1 Agroecosystem History Laboratory, University Pablo de Olavide, 41013, Seville, Spain; giguzcas@upo.es (G.I.G.); mgonnav@upo.es (M.G.d.M.)
2 CEAOAO & CEAC Tierra, Department of Animal Biology, Vegetal Biology and Ecology, University of Jaén, Jaen 23009, Spain; rgarcia@ujaen.es
3 CEIGRAM-ETSIAAB, Technical University of Madrid (UPM), Madrid, 28040, Spain; eduardo.aguilera@upm.es
* Correspondence: gcargal@upo.es

**Abstract:** Wheat yields are predicted to decrease over the next decades due to climate change (CC). Mediterranean regions are characterized by low soil fertility and stressful conditions that limit the effect of technological improvements on increasing yield gains, while worsening the negative CC impacts. Additionally, organic farming (OF) lacks specifically adapted genetic material. Accordingly, there is a need to search for varieties adapted to these conditions and whose cultivation may help semi-arid agroecosystems sustainability, focusing on specific agronomic and functional traits. To this purpose, wheat landraces and modern wheat varieties were evaluated under Mediterranean rainfed conditions during three growing seasons under contrasting situations: A conventional farm and an organic farm. Results regarding straw production, weed biomass and biodiversity, and grain N concentration suggest that the cultivation of landraces under Mediterranean rainfed conditions can enhance agroecosystem sustainability through positive effects on ecosystem services such as soil quality, functional biodiversity, or grain protein content, without significant reductions in grain yield. Results highlight the relevant role of wheat landraces as genetic resources for the development of cultivars adapted to Mediterranean agroecosystems conditions, especially for organic farming, but also for conventional agriculture.

**Keywords:** semi-arid; cereal landraces; ecosystem services; biodiversity; organic farming

## 1. Introduction

Wheat is grown in more than 220 million ha in the world, being the most widely cultivated crop [1]. Approximately 70% of global wheat cropland is under rainfed conditions [2]. In the Mediterranean basin, wheat is also mostly grown under rainfed conditions, and in Spain, this proportion reaches 86% of the total wheat area [3]. Worldwide, a decrease in cereal yields due to CC has been predicted [4], since the harmful climate effects can offset technological-based increases on cereal yields [5]. Mediterranean climate is characterized by low and erratic distribution of rainfall, along with growing temperature towards the end of the crop cycle. These conditions can reduce yield stability and worsen climatic impacts on cereal yields in the Mediterranean basin, where cereal yields are not mainly constrained by technology, but by environmental limits [6].

The increasing variability related to CC can call into question the success of modern varieties [7]. Particularly, under Mediterranean climate, water availability and low soil fertility are major factors determining wheat grain yield, and yield gains by genetic breeding strongly rely on environmental

ameliorations like irrigation and N fertilization [8]. CC is also threatening the provision of other ecosystem services besides yield [9], thus functional traits of cultivars should be also taken into account when evaluating the suitability of cultivars to be grown. Modern varieties likely lack clear adaptive traits for performing particularly well under stressful conditions [10]. In this context, landraces could constitute an alternative plant type better adapted to farming systems of less favorable areas [11], like semi-arid Mediterranean drylands [12].

Mediterranean wheat landraces harbor specific adaptations for low-input farming and tolerance to drought and heat stress [13]; therefore, they can help face new agricultural challenges in semi-arid agroecosystems. For example, their greater root systems might enhance water uptake from deep soil layers [12], making them more adapted to rainfed conditions. Likewise, improving N-use efficiency can help to enhance sustainability in semi-arid farming systems [14]. Larger roots of Mediterranean landraces can increase N uptake efficiency [15] due to the recovery of N from deeper soil layers [16]. The use of modern varieties under organic and low-input conditions could be associated with insufficient levels of protein for the milling industry [15], such as breeding programs selected for yield improvement with the trade-off of decreasing grain N [17].

The sustainability of production systems is also threatened by a high use of herbicides, which is negatively impacting crop fields biodiversity [18] and enhancing weed resistance expansion [19]. Weeds are a major concern for farmers, and finding more competitive crop varieties is important in both conventional and organic production systems [20]. It is widely accepted that cultivars grown before the expansion in the use of herbicides are more competitive against weeds [21], and the relevance of weeds biodiversity in Mediterranean cropland is widely acknowledged (e.g., Chamorro et al. [22]). Nonetheless, there is a research gap on the effect of landraces on weed biodiversity, particularly in relation to biomass production and under Mediterranean conditions.

Water scarcity affects semi-arid agroecosystem sustainability through the reduction of residues incorporation, with the concomitant negative impact on SOM levels [23]. Wheat varieties with higher straw production can contribute to the improvement of soil quality through residue incorporation [24] without diminishing straw output for the farmer.

Despite the fact that landraces can show some traits relevant for their agronomic and functional performance, to our knowledge, there is a lack of studies comparing wheat landraces and modern wheat cultivars traits under different managements and Mediterranean conditions. Our hypothesis is that wheat landraces cultivation under rainfed Mediterranean conditions can result in agronomic and functional benefits for the sustainability of both conventional and organic cropping systems. Hence, landraces and modern cultivars under organic and conventional management were evaluated in two separate field experiments that were carried out during three consecutive growing seasons (2013–2016), and grain yield, straw biomass, weed infestation, weed biodiversity, and N and C contents of wheat biomass were determined.

## 2. Materials and Methods

### 2.1. Experimental Design and Locations Description

Two field experiments with contrasting agronomic managements were run at two locations in the South of Spain (Figure 1)—Sierra de Yeguas (Málaga province) and La Zubia (Granada province)—during three consecutive years (2013–2016). The main soil properties are shown in Table 1, while mean annual temperature and annual rainfall are shown in Table 2. Mean annual temperatures during the experiment were similar to the mean values of the last 30 years for both locations. Nevertheless, mean annual rainfall was 56.4% and 68.9% of the 30-year average precipitation in Sierra de Yeguas and La Zubia, respectively.

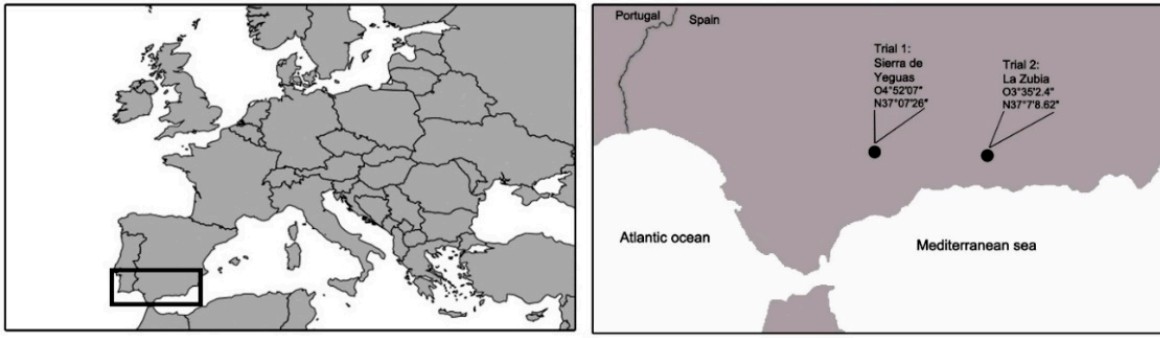

**Figure 1.** Trials locations in Southern Iberian Peninsula.

**Table 1.** Soil physico-chemical properties of the field experiments two weeks before the trials were run. Mean values and standard error are shown.

| Properties | Sierra de Yeguas | La Zubia |
|---|---|---|
| CEC (meq/100 g) | 31.19 ± 0.93 | 16.86 ± 0.85 |
| Ca exchangeable (meq/100 g) | 21.94 ± 0.79 | 13.83 ± 0.93 |
| Mg exchangeable (meq/100 g) | 5.80 ± 0.54 | 2.05 ± 0.72 |
| Na exchangeable (meq/100 g) | 1.34 ± 0.07 | 0.50 ± 0.07 |
| K exchangeable (meq/100 g) | 2.12 ± 0.05 | 0.48 ± 0.02 |
| Carbonate (%) | 12.27 ± 3.17 | 18.62 ± 0.20 |
| Limestone (%) | 4.61 ± 1.71 | 4.71 ± 0.43 |
| Olsen P (ppm) | 33.7 ± 4.4 | 27.0 ± 6.6 |
| SOC (%) | 1.39 ± 0.11 | 1.51 ± 0.17 |
| N org (%) | 0.16 ± 0.01 | 0.17 ± 0.01 |
| pH | 8.18 ± 0.02 | 7.99 ± 0.05 |
| pH in ClK | 7.46 ± 0.02 | 7.46 ± 0.03 |
| Assimilable K (ppm) | 927.0 ± 27.3 | 208.4 ± 6.2 |
| Clay (%) | 42.2 ± 1.1 | 16.4 ± 1.2 |
| Sand (%) | 18.6 ± 1.4 | 28.7 ± 3.5 |
| Silt (%) | 39.1 ± 0.9 | 54.8 ± 2.4 |
| Texture | Clay | Silt-loam |

CEC = cation exchange capacity; SOC = soil organic carbon.

The terrain at Sierra de Yeguas had been under organic (ORG) farming conditions (Table 2) for 15 years. Wheat and faba bean (Vicia faba) were grown in rotation. Poultry manure was applied to wheat before seeding, but not to faba bean. Weeds were cursory controlled by hand in February. The terrain at La Zubia was characterized by a monoculture wheat cropping system (CON) with synthetic fertilizers and herbicide weed control. Synthetic fertilizer was spread before seeding, while weeds were controlled with a broad-leaf herbicide, applied before stem elongation. Wheat was the main crop cultivated at both sites before the trial.

Both farmlands were seeded between 25 October and 12 November. Sowing rates were 200 kg ha$^{-1}$ and 110 kg ha$^{-1}$ for wheat and faba bean, respectively. The harvest of wheat and faba bean was done between 5 June and 25 June.

The experimental design was a randomized complete block with four blocks separated by a non-cultivated stripe 1 m width. Plot size was 4 × 6 m$^2$. Both phases of the ORG rotation (wheat and faba) grew in adjacent plots and interchanged cultivation plots each cropping season.

**Table 2.** Climatic characteristics and main management practices of the field experiments.

| | ORG | CON |
|---|---|---|
| *Location* | Sierra de Yeguas | La Zubia |
| *Mean annual temperature (°C)* | | |
| *2013–2014* | 16.0 | 15.3 |
| *2014–2015* | 16.6 | 17.0 |
| *2015–2016* | 16.9 | 15.4 |
| *1982–2012 average* | 16.3 | 15.2 |
| *Annual precipitation (mm)* | | |
| *2013–2014* | 433 | 309 |
| *2014–2015* | 344 | 359 |
| *2015–2016* | 363 | 288 |
| *1982–2012 average* | 673 | 462 |
| *Farming system* | Organic | Conventional |
| *Rotation* | Wheat-Faba bean | Monoculture |
| *Fertilization* | Poultry manure (3.6% N, d.m.) (3.0 Mg ha$^{-1}$, f.m.) | NPK (8:15:15) (570 kg ha$^{-1}$) |
| *N (kg ha$^{-1}$)* | 54 (wheat) + 27 (faba bean) | 45.6 |
| *P$_2$O$_5$ (kg ha$^{-1}$)* | n.d. | 85.5 |
| *K$_2$O (kg ha$^{-1}$)* | n.d. | 85.5 |
| *Weed control* | Manual weeding | MCPA 40% (2 l ha$^{-1}$) |
| *Irrigation* | Rainfed | Rainfed |

f.m. and d.m. stand for fresh and dry matter, respectively, while n.d. stands for not determined.

### 2.2. Plant Material

Six landraces (Rubio, Recio, Sierra Nevada, Barbilla Roja, Rojo Pelón, and Blanco Verdial) (Table S1) and six modern cultivars (Avispa, Simeto, Vitrón, García, Marius, and Artur Nick) were seeded at both farmlands. The seeds of the landraces came from the Phytogenetic Resource Centre of the National Agrarian Research Institute of Spain (CRF-INIA), and they were varieties commonly cultivated in the region during the first third of the 20th century. Modern varieties were selected among currently used varieties, taking into consideration their good reputation among farmers in the area.

### 2.3. Agronomic Variables

Aerial net primary productivity (NPP), including total crop dry matter (CDM) and total weed dry matter; grain yield; straw biomass production; and weed biomass were sampled in $0.5 \times 0.5$ m$^2$ squares randomly arranged at the center of the plots. Samples of wheat and weed biomass were taken by cutting plants at ground level. Wheat plants were then separated into spike and stem, and fresh spikes were threshed to separate grain and grain husk. Wheat and weed biomass were dried at 70 °C to obtain dry weight. Harvest index (HI) and the ratio between weed biomass and NPP (weed:NPP), that is, the ratio of agroecosystem biomass allocated to weed, were also calculated.

For the total N and C analyses, dried samples of grain, straw, and weed biomass were milled (<1 mm) and analyzed with an elemental autoanalyzer CNOH-S (Flash EA1112 CHNS-O, Thermo Finnigan).

### 2.4. Weed Biodiversity Indices

Weed density and weed species were determined in 0.5x0.5 m$^2$ squares randomly arranged at the center of the plots during the flowering stage of most weeds in the Mediterranean area (April–May) of the second and third growing seasons. First, species richness index (S) was calculated, which measures the total number of species from the sample. Then, we calculated the following biodiversity indices. The Margalef index (1) determine species number in terms of numbers of individuals, and is not able to distinguish different levels of biodiversity between communities with the same number of

species and individuals since it ignores other components of biodiversity such as species evenness and community structure. In order to accomplish a more comprehensive analysis of biodiversity, we added the Shannon–Wiener index (2), which links biodiversity with higher uncertainty of randomly choosing an individual of a certain species. Then, Simpson (3) and Pielou (4) dominance and evenness indices, respectively, were included.

    (1) $D = (S - 1)/\ln N$,

where S is the number of species in the sample, and N is the specimen's total number,

    (2) $H' = -\Sigma((p_i)(\ln p_i))$,

    (3) $D = 1 - \Sigma p_i^2$,

    (4) $J' = H'/\ln S$,

where $p_i$ represents relative abundance of the species i (calculated as $n_i/N$, where $n_i$ is the number of individuals of the species i, and N is the total of individual from the sample).

### 2.5. Statistical Analysis

A complete randomized block variance analysis of variables measured in the fields was carried out within each year at a significance level of 0.05. Treatment means were compared using Tukey 's test at 0.05 probability level. Previously, a Shapiro–Wilk's normality test was run to check for normal distribution. Statistix software (Analytical Software, Tallahassee, FL, USA, Version 10) was run for every test.

Statistical comparisons between both farming systems (ORG and CON) were not performed, since site characteristics such as soil properties, water balances, etc., could have a relevant influence on the studied variables beyond management.

## 3. Results

### 3.1. Net Primary Productivity, Crop Dry Matter, and Weed Biomass

Total aerial NPP (Table S2), which includes CDM and weeds dry matter, ranged from 5957 kg ha$^{-1}$ to 19,158 kg ha$^{-1}$ produced by modern wheat under ORG management in 2016 and under CON in 2014, respectively. Under ORG management, NPP in plots with wheat landraces was 27% (2014) and 18% (2016) higher than that of plots with modern varieties, while no significant differences were recorded in 2015 (Figure 2). CDM followed the same pattern, with landraces out-yielding modern varieties in 2014 and 2016 (68% and 21% higher CDM, respectively). Mean three-growing-season grain yields for landraces and modern varieties under ORG were 2028 kg ha$^{-1}$ and 2201 kg ha$^{-1}$. While significant differences for grain yield were only found in 2014 and 2015, with 79% higher and 36% lower yield for old wheat compared to modern one, respectively, straw production was always higher for landraces. Landraces straw biomass was 62%, 31%, and 30% higher than that of modern cultivars in 2014, 2015, and 2016, respectively. However, husk residues from wheat were 86% higher for landraces in 2014, and no differences were found the rest of the years. HI ranged 0.211–0.238 and 0.393–0.206 for landraces and modern varieties, respectively. No significant differences were found in 2014 and 2016, while it was 39% significantly lower for landraces in 2015 (Table 3).

Under CON management, NPP of landraces was 35% significantly higher than that of modern varieties in 2015, while no significant differences were found in the rest of the years (Figure 2). Contrastingly, CDM of old wheat was significantly higher in 2015 and 2016 (49% and 18% higher, respectively). Mean three-growing-season grain yields of landraces and modern wheat cultivars were 1824 kg ha$^{-1}$ and 2054 kg ha$^{-1}$, respectively. Significant differences for grain yield were only found in 2014, when old wheat had 27% lower yield than modern wheat. Straw residues productions were not different in 2014 and 2016, while it was 61% higher for landraces in 2015. Regarding husk residues, significant differences between varieties were detected only in 2016, when landraces produced 36% higher husk biomass. HI ranged 0.105–0.145 and 0.125–0.179 for old and modern cultivars,

respectively. Significant differences were found in 2014 and 2015, with 25% and 34% lower HI for landraces, respectively.

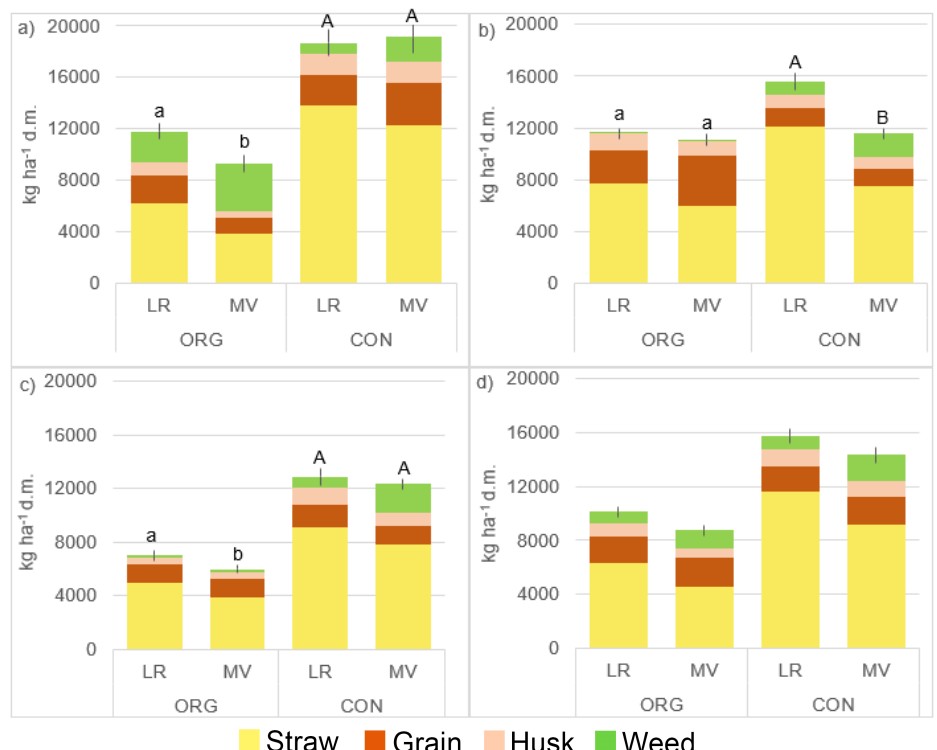

**Figure 2.** Grain yield, straw biomass, and husk residue (kg ha$^{-1}$ dry matter) produced by landraces (LR) and modern wheat varieties (MV) and weed biomass in 2014 (**a**), 2015 (**b**), 2016 (**c**), and the three-year average (**d**). Aerial net primary productivity corresponds to bars total height. LR = landraces; MV = modern varieties. Different letters within each location indicate significant differences between LR and MV (ANOVA, $p < 0.05$). Bars stand for standard error of the mean ($n = 24$).

**Table 3.** Harvest index (HI) (fresh matter), grain N uptake, and weed:NPP (net primary productivity) ratio (aerial dry matter) of landraces (LR) and modern wheat varieties (MV) in organic (ORG) and conventional (CON) trials. Mean and standard error of the mean ($n = 24$). Different letters represent significant differences between LR and MV within each growing season at a level of 0.05 (Tukey test).

|  |  | 2014 | | 2015 | | 2016 | |
|---|---|---|---|---|---|---|---|
|  |  | **LR** | **MV** | **LR** | **MV** | **LR** | **MV** |
| **HI** | **ORG** | 0.235a ± 0.02 | 0.206a ± 0.03 | 0.238b ± 0.01 | 0.393a ± 0.01 | 0.211a ± 0.02 | 0.262a ± 0.02 |
|  | **CON** | 0.133b ± 0.01 | 0.179a ± 0.02 | 0.105b ± 0.01 | 0.158a ± 0.03 | 0.145a ± 0.02 | 0.125a ± 0.02 |
| **Grain N** | **ORG** | 43.4a ± 5.0 | 19.7b ± 2.9 | 66.5b ± 2.6 | 87.8a ± 5.3 | 31.3a ± 2.9 | 27.9a ± 2.5 |
| **uptake** | **CON** | 72.8a ± 5.2 | 79.3a ± 6.9 | 29.1a ± 3.5 | 24.1a ± 4.1 | 34.8a ± 3.3 | 26.2b ± 2.9 |
| **Weed:NPP** | **ORG** | 0.210b ± 0.04 | 0.371a ± 0.04 | 0.0003b ± 0.00 | 0.004a ± 0.00 | 0.019b ± 0.00 | 0.041a ± 0.01 |
|  | **CON** | 0.049b ± 0.01 | 0.101a ± 0.01 | 0.064b ± 0.01 | 0.159a ± 0.02 | 0.066b ± 0.02 | 0.181a ± 0.03 |

Plots cultivated with landraces had significantly lower weed biomass under both managements (Figure 2). This trend was also observed for the contribution of weed biomass to total NPP (weed:NPP ratio). Under ORG management, plots underwent a very high weed infestation in 2014, and weed biomass in plots cultivated with landraces was 35%, 91%, and 50% lower than those of modern varieties in 2014, 2015, and 2016, respectively. Accordingly, landraces had lower weed:NPP ratio values than modern varieties (44%, 93%, and 55% lower values in 2014, 2015, and 2016, respectively). Under CON management, weed biomass and weed:NPP for landraces were also significantly lower than those of modern varieties. Values for the former were 45%–61% and 52%–64% significantly lower than for the latter for weed biomass and weed:NPP ratio, respectively.

## 3.2. Weed Density and Biodiversity

Weed density ranged from 9.1 to 140.7 plants m$^{-2}$ (Table 4). Significant differences were found for both years and managements, with landraces plots harboring a lower weed density than modern ones (40% and 64% lower weed abundance under ORG and CON, respectively).

**Table 4.** Richness, density (plants m$^{-2}$), and biodiversity indices of weed community of plots planted with landraces (LR) and modern varieties (MV) under ORG and CON management in 2015 and 2016. Mean values and standard error of the mean (*n* = 24). Different letters represent significant differences between LR and MV within each growing season at a level of 0.05 (Tukey test).

| | | | Richness | Density | Margalef | Simpson | Shannon | Pielou |
|---|---|---|---|---|---|---|---|---|
| **2015** | ORG | LR | 1.79b ± 0.19 | 9.08b ± 1.35 | 0.73a ± 0.12 | 0.37a ± 0.05 | 0.58a ± 0.08 | 0.71a ± 0.08 |
| | | MV | 2.63a ± 0.21 | 23.05a ± 3.32 | 0.71a ± 0.10 | 0.39a ± 0.04 | 0.66a ± 0.08 | 0.64a ± 0.07 |
| | CON | LR | 3.92b ± 0.29 | 26.39b ± 3.59 | 1.26a ± 0.09 | 0.61a ± 0.03 | 1.14a ± 0.07 | 0.83a ± 0.02 |
| | | MV | 5.54a ± 0.45 | 91.45a ± 10.79 | 1.23a ± 0.12 | 0.53a ± 0.04 | 1.09a ± 0.09 | 0.64b ± 0.04 |
| **2016** | ORG | LR | 3.71b ± 0.23 | 89.33b ± 11.86 | 0.77b ± 0.06 | 0.43a ± 0.03 | 0.77a ± 0.06 | 0.62a ± 0.04 |
| | | MV | 5.09a ± 0.44 | 140.70a ± 13.55 | 1.00a ± 0.11 | 0.42a ± 0.04 | 0.81a ± 0.09 | 0.50b ± 0.04 |
| | CON | LR | 5.08a ± 0.43 | 50.35b ± 5.58 | 1.48a ± 0.18 | 0.61a ± 0.04 | 1.20a ± 0.09 | 0.77a ± 0.04 |
| | | MV | 5.71a ± 0.46 | 124.90a ± 10.78 | 1.15a ± 0.11 | 0.55a ± 0.04 | 1.10a ± 0.09 | 0.63b ± 0.03 |

Richness of weed species ranged from 1.79 to 5.71, except in 2016 under CON management. Overall, weed richness was 29% and 20% lower for landraces in ORG and CON, respectively.

The rest of biodiversity indices measured followed a different pattern. The Margalef index ranged from 0.71 to 1.48. Significant differences were only found in 2016 in ORG trial, when index values were 0.77 and 1 for landraces and modern wheat, respectively. Regarding Simpson and Shannon indexes, significant differences were not found. Finally, the Pielou index was significantly higher for landraces under ORG (2016) and CON (2015 and 2016).

Under ORG management, *Stellaria media* and *Melilotus indicus* were the most abundant weed species (Table S3) and their densities were 47% and 77% lower for landraces, respectively, in 2015 (*p* = 0.0007 and *p* = 0.0090, for *S. media* and *M. indicus* density reduction, respectively). In 2016, *S. media* density was 39% lower for landraces (*p* = 0.0007). Under CON management, *Lolium rigidum* was clearly dominant (Table S4), and the cultivation of landraces significantly reduced its density by 75% and 74% in 2015 and 2016, respectively (*p* = 0.0000 for both years). In addition, a significant reduction of 69% was also found for the second dominant weed species, *Phalaris minor* (*p* = 0.0132), in 2015.

## 3.3. Grain Nitrogen and Straw Carbon Content

Grain N concentrations of landraces were significantly higher across the three years of the experiment and under both managements. Averaged percentages were 2.35% and 2.0% for landraces and modern wheat under ORG, and 2.52% and 2.18% under CON, respectively.

Grain N uptake ranged 19.7–87.8 kg ha$^{-1}$ under ORG and 24.1–79.3 kg ha$^{-1}$ under CON (Table 3). Landraces showed similar or higher values than modern varieties, except in 2015 under ORG, when modern varieties significantly out-yielded old ones.

Straw C concentration of landraces was significantly higher than that of modern varieties in 2015 and 2016 and both managements (Figure 3). Across the years, averaged values were 41.8% and 41.4% for ORG management, and 41.7% and 40.5% for CON, respectively, for old and modern cultivars.

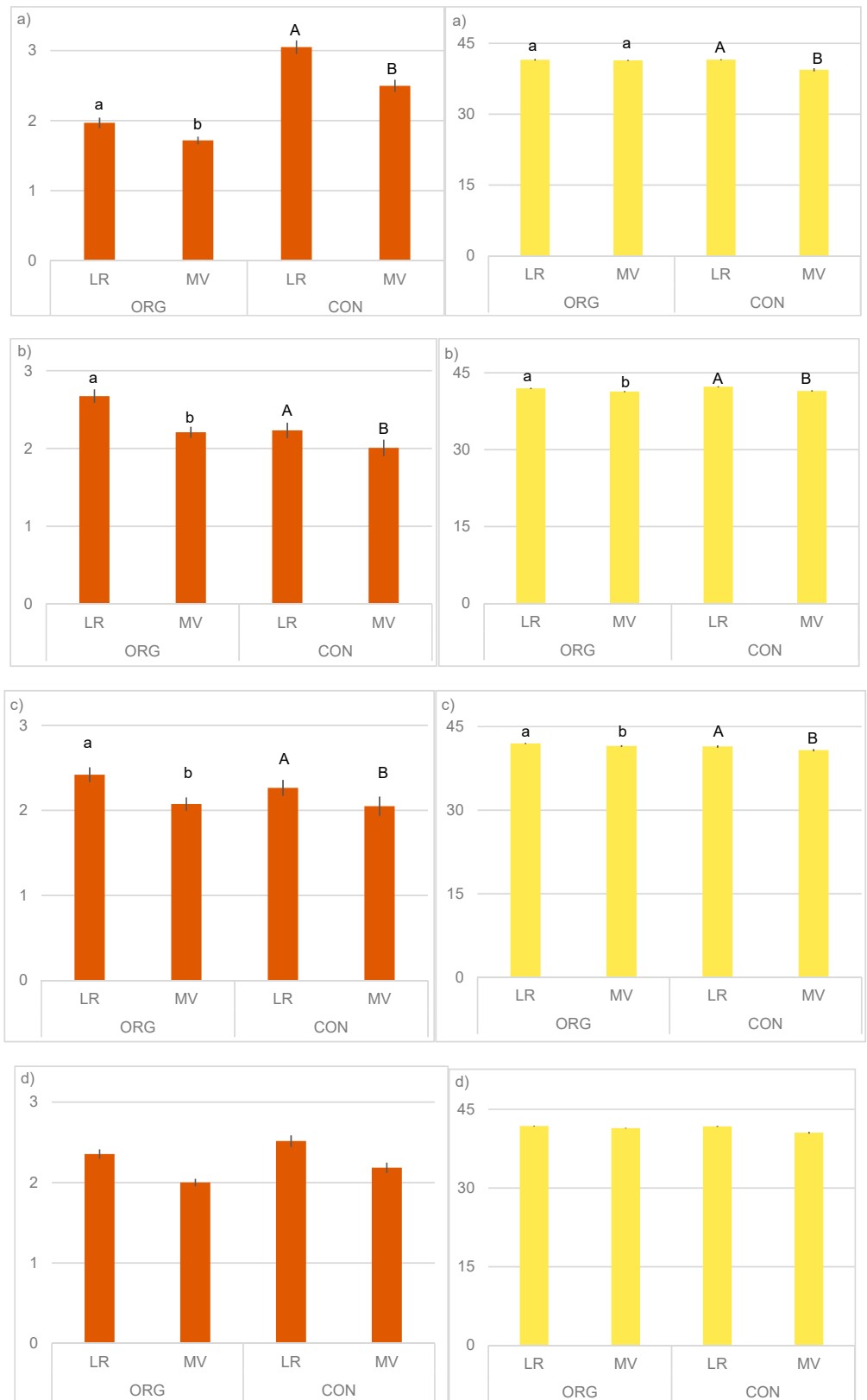

**Figure 3.** Nitrogen grain (left figures) and straw carbon (right figures) content (%) of landraces (LR) and modern varieties (MV) grown under ORG and CON management in 2014 (**a**), 2015 (**b**), 2016 (**c**), and three-year average values (**d**). Different letters within each location indicate significant differences between LR and MV (ANOVA, *p* < 0.05). Bars stand for standard error of the mean (*n* = 24).

## 4. Discussion

### 4.1. Crop Dry Matter, Grain Yield, and Straw Biomass

There is a lack of consensus on differences in CDM between landraces and modern cultivars. Across the years and growing seasons, CDM of landraces was higher than that of modern wheat, as observed by Ayadi et al. [25]. Contrastingly, De Vita et al. [26] and Royo et al. [27] found no significant differences for CDM among a set of cultivars of different year of release in Italy and Spain, respectively. These authors found higher grain yield for lately released cultivars, likely due to a higher fertilization dose than in the present study and to the application of irrigation, respectively. This fact led to a compensation for the lower straw biomass production found for modern varieties. However, our results showed no significant differences in grain yield, and therefore the higher straw biomass production was the main factor responsible for the higher CDM of landraces. Other studies have reported higher straw biomass production of landraces [28], which can have potential sustainability benefits. Cereal straw mulching has been proven to effectively reduce soil erosion and increase soil quality, biological activity, and soil aggregate stability [29] under Mediterranean rainfed conditions. In this sense, higher straw production of old cereal has been valued as a relevant source of organic material for increasing soil organic matter [30]. The increase of soil organic matter can contribute to enhance yield without significant increases in soil GHG emissions under semi-arid conditions [31] and prevent soil structure degradation [29]. In addition, crop residue incorporation contributes to ameliorate the predicted yield losses under CC scenarios in semi-arid areas by improving nutrient availability [32] and increasing cereal's water-use efficiency [33].

Overall, grain yield levels from this study are within typical ranges of rainfed wheat under conventional management in Spain for the growing seasons of the experiment (2665 kg ha$^{-1}$) [34,35]. The higher grain yield of modern wheat cultivars [25,36] is widely accepted. Contrastingly, our result showed similar grain yields of landraces and modern varieties under the same N fertilization. Under ORG, the lack of yield differences could be partially due to the breeding process, which would have led to modern varieties to be more dependent on easy access to nutrients [16] and on the presence of herbicides [37]. Studies commonly focus on maximizing the grain yield response to N supply, instead of maintaining or increasing grain yields at a given rate of available N [38]. Our results under ORG management suggest that modern wheat could not out-yield old wheat under circumstances of relatively low weed control, available nutrients, and water. Similarly, Murphy et al. [21] found that some of the landraces tested under organic fertilization out-yielded modern cultivars under higher precipitation conditions (500 mm/year).

The lack of differences under CON management disagree with other studies under Mediterranean climate [36]. This could be due to the specific edaphoclimatic conditions of our experiment. Under Mediterranean conditions, rainfed crops are highly dependent on rainfall. Thus, it is likely that the similar yields found for both varieties could be due to the low annual rainfall during the experiment (Table 2), and this could have also influenced ORG outputs. These results are in line with studies showing that yield gains achievements in modern wheat cultivars during the last century can be under-expressed if wheat is grown under stressful conditions like semi-arid rainfed conditions [27].

In addition, weed biomass [39] and, particularly, *Lolium* dry matter [40], have been found to be responsible for grain yield reductions in modern varieties. In Mediterranean drylands, the presence of weeds, along with rainfall, is considered a major factor affecting grain yield [41].

The HI of landraces grown under ORG are slightly lower than those reported for other Spanish landraces in Royo et al. [27], while values of modern varieties under CON were lower than common references for modern cultivars (e.g., Ayadi et al. [25]). This could be due to the fact that water deficits in rainfed environments can lead to reduced HI [42] and to the relatively low annual N inputs of our experiment, since HI is highly influenced by N availability [25].

### 4.2. Weed Biomass and Weed:NPP Ratio

Data from this study reveal that landraces were better competitors against weeds, as they produced lower weed biomass in both absolute and relative terms. This result agrees well with many studies [40], also under organic [21,39] and inorganic fertilization [30]. The competitive ability of cultivars grown before the expansion of herbicide application is related to increased height [21], aboveground biomass and leaf area index [39], and higher competition for N availability [30]. In addition, some authors have related weed suppression ability with the presence of allelopathic compounds in these older genotypes [43], and analyses of the cultivars from this study show the same trend [44].

Our results suggest that the cultivation of landraces can mitigate the requirements of herbicides; therefore, these genetic resources can contribute to the sustainability of farming systems and to the reduction of the environmental damage related to the use of these chemicals. Under organic or low input farming conditions, weeds constitute a major factor of yield reductions [45], and cultivars with higher weed competition are required [7]. Landraces under Mediterranean conditions can help to keep weeds under acceptable thresholds, reducing the weed seed bank in the soil for the next crop [30]. This is of special relevance under organic farming conditions, where herbicide use is forbidden.

### 4.3. Weed Density and Biodiversity

Weed density and species number in plots cultivated with modern wheat cultivars were higher than in plots with old wheat cultivars. Commonly, a higher weed density is found together with a higher richness index [46]. Among biodiversity indexes, only the Pielou index differed significantly between plots with different varieties, with higher values for landraces plots. Therefore, richness index seemed to be insufficient to measure biodiversity weed population in our study. Maintaining biodiversity within field crops is an important factor for sustainable agriculture [47]. These results suggest that the cultivation of landraces could reduce weed density while maintaining (or increasing) the diversity within weed community and the crop yield, which is a growing concern in Mediterranean cereal fields (e.g., Armengot et al. [41]). Organic management has been proven to increase functional weed biodiversity [22], and our results indicate that the cultivation landraces in organically managed Mediterranean drylands can help keeping weeds controlled while preserving ecosystem services related to biodiversity. Regarding conventional management conditions, wheat landraces could help in reducing herbicide application and its negative effects. Herbicide is one of the agricultural intensification factors behind the depletion of weed functional biodiversity in Mediterranean fields [22], negatively affecting the trophic web and other agronomic ecosystem services like pollination, pest control, and nutrient cycling [18]. In addition, herbicide application can enhance soil erosion by decreasing the vegetation cover and preventing soil development [29].

### *Lolium* sp.

*Lolium rigidum* was the weed dominating infestation in the conventional trial. Accordingly, Romero et al. [48] reported *L. rigidum* as the dominant weed in 60% of the cases in their study. Landraces suffered from a lower *Lolium* infestation than modern varieties under CON, in agreement with previous findings [40].

*Lolium* sp. is considered the first widespread herbicide resistant weed in the world [37], and it affects croplands of many countries [19]. It has been related to land-use intensity, and increased in importance within cereal fields since the 1970s [49]. In Spain, the observed shift from broad-leaved to grass weeds could be due to the use of broad-leaf herbicides [49] and the appearance of herbicide resistance [48].

Results from this study show that landraces can contribute to reduce *Lolium* infestation under CON management, and its negative impact on cereal yields. This functional trait could help to broaden the options to decrease this relevant grass infestation under Mediterranean conditions. Future collaborations among cereal farmers and researchers should aim to develop successful

strategies to reduce this problem through the cultivation and/or the development of cultivars adapted to Mediterranean agroecosystems from these landraces.

### 4.4. Grain Nitrogen and Straw Carbon Content

Our results for grain N content are in the range of those by Ercoli et al. [50] in a Mediterranean region, and slightly higher than those from Dawson et al. [38] under organic fertilization. According to our results, landraces harbor higher grain N concentration than modern varieties, in agreement with previous authors [38,51,52]. Accordingly, Lazzaro et al. [39] found higher grain protein content in traditional wheat cultivars than in modern ones in their organic trial under Mediterranean conditions. The same pattern was found for different N fertilizer rates [26,36] and different interspecific competition levels [30].

Usually, selection for high yields reduced N in grain, with a negative correlation between grain N content and yield being widely reported in modern varieties [17,30]. Nevertheless, according to our results, modern varieties were not able to counterweight low yields with a higher grain N content under low-fertility conditions, when comparing to old ones. Therefore, under the same rainfed Mediterranean conditions and soil N availability, landraces seemed to have a higher ability to concentrate N in grain, and to achieve higher or similar values of grain N harvested (except in ORG-2015) without jeopardizing grain yields.

Protein content is an essential trait for the end-use quality of wheat [26]. Landraces from this study thus show an important advantage particularly for organic farming, as organic wheat often shows low protein contents [45]. The need for improvement of the grain protein content of organic wheat has been highlighted to better match the food industry requirements [15], and landraces could be a plausible alternative.

Landraces allow higher availability of C inputs to be incorporated into the soil after the harvest without compromising other straw uses to meet farmer necessities. Higher C inputs can lead to a higher SOC sequestration [53]; thus, high-residue-yielding varieties could contribute to reduce SOC depletion vulnerability under these climate conditions [54]. Moreover, SOC increases can offset a relevant share of agricultural GHGe [55], contributing to CC mitigation. Recently, a life-cycle assessment of old and modern wheat cultivars in this experiment concluded that higher C input due to a higher aboveground and belowground residue production of landraces would lead to a significant reduction in the C footprint of wheat landraces through soil C sequestration [56]. These results point out the potential of wheat landraces as a CC mitigation strategy in rainfed Mediterranean agroecosystems.

Genetic breeding in cereal crops is responsible for varieties able to produce higher yields in monoculture cropping systems [57]. Trade-offs of this improvement impacted weed competition ability [37], straw production [26], and grain protein content and quality [17,26]. In addition, modern breeding contributed to eroding the genetic diversity of staple cereals [58]. However, in this study, we show that modern cereal cultivars could not perform their yield advantage under rainfed Mediterranean conditions, thus jeopardizing agroecosystem sustainability due to higher requirements of chemical inputs and the worsening of soil quality, without a grain production benefit. Sener et al. [59] evaluated the genetic improvement of wheat during the last decades of the 1900s under Mediterranean conditions, showing a lower ability for improving yields compared to other regions due to specificities of Mediterranean climate. Low rainfall is partially responsible for the lower success of breeding under these conditions [10].

There is a growing interest in enhancing ecosystem services while maintaining or increasing yield [60]. For example, there is a need to take into account other indicators, like the quality of grain and the number of varieties per crop, on the way to sustainable agriculture [58]. The expansion of modern cereal under these edaphoclimatic conditions could be partially responsible for the loss of agricultural sustainability, therefore yield and sustainability performance evaluations of landraces under these conditions should be attracting research attention.

Previous studies have suggested the suitability of landraces under Mediterranean semi-arid conditions (e.g., Giambalvo et al. [30]), which can be relevant sources for sustainable agriculture in a CC context more suitable to adapt to future scenarios [61]. By cultivating landraces, we could also meet some sustainable crop intensification (SCI) practices described for a climate-resilient and sustainable agriculture [62], like the use of plants with vigorous early growth [63] and reducing the needs of herbicide application. This research contributes to a comprehensive approach for the study of landraces under rainfed Mediterranean conditions, showing the benefits of growing these plant genetic resources from an agronomic perspective (maintaining yields while increasing protein content) but also from a sustainability approach (contributing to soil quality, weeds reduction, and cropland biodiversity). Farmers that still grow landraces not only have an agronomic criterion, but socio-economic and geographical factors, along with farm characteristics, are influencing farmer decisions [64]. The research in landraces is important to contribute to a more solid decision-making by farmers, mostly from marginal areas or croplands with a low yielding potential. The present findings can help widen this recognized utility of landraces, highlighting some functional traits that could contribute to the development of a more sustainable agriculture, including the use of these landraces for the development of cultivars with those benefits.

## 5. Conclusions

The findings from this research unveil agronomic and functional advantages of landraces cultivation in Mediterranean rainfed fields that could have been overlooked in breeding programs searching for highly responsive varieties to technological improvements. Landraces from this study have proved to be more competitive against weeds, even against a relevant pest of cereal fields such as *Lolium*, allowing for a reduction in the need for herbicide spraying in conventional agriculture and broadening the weed control options of organic farming. In addition, the higher straw biomass production can contribute to CC mitigation, along with soil quality improvement. Furthermore, old cultivars were more efficient in N accumulation in grain. Relevantly, all of this was possible without decreasing grain yields and maintaining or increasing weed community biodiversity indexes. These results emphasize the role of landraces as plant genetic resources that can improve the sustainability of Mediterranean rainfed agroecosystems.

**Supplementary Materials:** The following are available online at http://www.mdpi.com/2071-1050/11/21/6029/s1, Table S1: Phenological data of landraces, Table S2: Grain yield, straw, husk, and weeds biomass production of landraces and modern wheat varieties; Table S3: Weed species density in Sierra de Yeguas (ORG); Table S4. Weed species density in La Zubia (CON).

**Author Contributions:** Conceptualization, G.I.G.; data curation, G.C.-G. and G.I.G.; formal analysis, G.C.-G. and G.I.G.; funding acquisition, M.G.d.M.; investigation, G.C.-G. and G.I.G.; methodology, G.C.-G., G.I.G., and R.G.-R.; project administration, M.G.M.; resources, G.C.-G., G.I.G., R.G.-R., E.A., and M.G.M.; supervision, G.C.-G., G.I.G., R.G.-R., E.A, and M.G.M.; validation, G.C.-G. and G.I.G.; visualization, G.C.-G., G.I.G., E.A., and M.G.M.; writing—original draft, G.C.-G. and G.I.G.; writing—review and editing, G.C.-G., G.I.G., and E.A.

**Funding:** This work was supported by the international research project SSHRC 895-2011-1020 granted by the Social Sciences and Humanities Research Council of Canada, and the Spanish research projects HAR2012-38920-C02-01 and HAR2015-69620-C2-1-P granted by the Spanish Ministry of Economy and Competitiveness. The first authoress held a FPU scholarship from the Spanish Government. E.A. is funded by a Juan de la Cierva research contract from the Ministerio de Economía y Competitividad of Spain (FJCI-2017-34077).

**Conflicts of Interest:** The authors declare no conflict of interest. The funders had no role in the design of the study; in the collection, analyses, or interpretation of data; in the writing of the manuscript, or in the decision to publish the results.

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
