# Peer review of "Addressing the Role of Landraces in the Sustainability of Mediterranean Agroecosystems"

_sustainability, doi:10.3390/su11216029_

Round 1

Reviewer 1 Report

The paper made by Guiomar Carranza-Gallego et al. is very relevant focusing on a topic of enormous interest from the agro-ecological point of view. The study is well designed and the parameters analyzed well selected. I have some specific comments related to the landraces and modern varieties used, and some remarks related to the results presented.

Section 2.2.

I suggest to include the basic traits of the six old landraces and the six modern varieties of winter wheat in order to provide information about the height, aboveground biomass and tillering performance; at least the range of variation between them. The authors indicate that the seeds of the old landmarks came from the CRF-INIA. Did the authors cultivate plants from the accessions to obtain enough seeds? Did the CRF-INIA provide all seeds?

Section 2.3. The harvest index (HI) is not included in this section, but it is mentioned in the table 3.

Section 2.4. I suggest reducing the number of indexes of diversity, or alternatively explain clearly the benefits of the study is to use simultaneously the Margalef and the Shannon-Wiener indexes, and clarify that Simpson and Below are the indexes of dominance and evenness respectively.  Furthermore, I suggest to review critically the section 3.2 because is extremely descriptive and redundant.  

 Section 3. Results

I suggest to include information about the performance of the different varieties when are growing in the ORG and the CON fields. This information is relevant in order to understand the variability of results.

Section 3.2. Rewrite this section, see previous comments.

Figure 3. Include the standard error In the bars.

References.  The number of references is extremely high. I suggest authors review the paper trying to reduce the number of references.  

The paper made by Guiomar Carranza-Gallego et al. is very relevant focusing on a topic of enormous interest from the agro-ecological point of view. The study is well designed and the parameters analyzed well selected. I have some specific comments related to the landraces and modern varieties used, and some remarks related to the results presented.

Section 2.2.

I suggest to include the basic traits of the six old landraces and the six modern varieties of winter wheat in order to provide information about the height, aboveground biomass and tillering performance; at least the range of variation between them. The authors indicate that the seeds of the old landmarks came from the CRF-INIA. Did the authors cultivate plants from the accessions to obtain enough seeds? Did the CRF-INIA provide all seeds?

Section 2.3. The harvest index (HI) is not included in this section, but it is mentioned in the table 3.

Section 2.4. I suggest reducing the number of indexes of diversity, or alternatively explain clearly the benefits of the study is to use simultaneously the Margalef and the Shannon-Wiener indexes, and clarify that Simpson and Below are the indexes of dominance and evenness respectively.  Furthermore, I suggest to review critically the section 3.2 because is extremely descriptive and redundant.  

 Section 3. Results

I suggest to include information about the performance of the different varieties when are growing in the ORG and the CON fields. This information is relevant in order to understand the variability of results.

Section 3.2. Rewrite this section, see previous comments.

Figure 3. Include the standard error In the bars.

References.  The number of references is extremely high. I suggest authors review the paper trying to reduce the number of references.  

Author Response

The paper made by Guiomar Carranza-Gallego et al. is very relevant focusing on a topic of enormous interest from the agro-ecological point of view. The study is well designed and the parameters analyzed well selected. I have some specific comments related to the landraces and modern varieties used, and some remarks related to the results presented.

 Section 2.2.

Point 1: I suggest to include the basic traits of the six old landraces and the six modern varieties of winter wheat in order to provide information about the height, aboveground biomass and tillering performance; at least the range of variation between them. The authors indicate that the seeds of the old landmarks came from the CRF-INIA. Did the authors cultivate plants from the accessions to obtain enough seeds? Did the CRF-INIA provide all seeds?

Response 1: Regarding basic traits of cultivars, aboveground biomass is part of the results, so we are not sure of the convenience of including it in this section. With respect to the rest of the requested data, we have included a table with the phenological variables of landraces available at CRF web site and from our field experiments in the supplementary data.

CRF-INIA provided enough seeds to start the field experiment the first year. For the following years, seeds used to sow the fields were those recollected by hand during the harvest of previous season.

Point 2: Section 2.3. The harvest index (HI) is not included in this section, but it is mentioned in the table 3.

Response 2: Mention to HI has been included in line 152.

Point 3: Section 2.4. I suggest reducing the number of indexes of diversity, or alternatively explain clearly the benefits of the study is to use simultaneously the Margalef and the Shannon-Wiener indexes, and clarify that Simpson and Below are the indexes of dominance and evenness respectively.  

Response 3: We have tried to follow the alternative that the reviewer offered as follows:

Line 162: “Then we calculated the following biodiversity indices. Margalef index (1), determine species number in terms of numbers of individuals, not being able to distinguish different levels of biodiversity between communities with the same number of species and individuals since it ignores other components of biodiversity such as species evenness and community structure. In order to accomplish a more comprehensive analysis of biodiversity, we added the Shannon-Wiener index (2), which links biodiversity with higher uncertainty of choosing randomly an individual of a certain species. Then, Simpson (3) and Pielou (4) dominance and evenness indices, respectively, were included”.

Point 4: Furthermore, I suggest to review critically the section 3.2 because is extremely descriptive and redundant.

Response 4: We agree with the reviewer and we have tried to summarize and make a clearer and more concise section.  

Point 5: Section 3. Results

I suggest to include information about the performance of the different varieties when are growing in the ORG and the CON fields. This information is relevant in order to understand the variability of results.

Response 5: We suppose the reviewer refers to statistical comparison between organic and conventional experiments. We did not compare organic and conventional on purpose, since our major aim was to test the potential role of genetic material of wheat landraces to improve rainfed agroecosystems performance and sustainability. We believe that the innovative insight of our contribution is the suitability of old wheat varieties for this specific objective and under these specific climatic conditions. Their appropriateness should not be shadowed behind the farming system considered, but enhanced after the comparison with modern wheat cultivars within each contrasting farming system. For that reason, we stablished the experiments in sites with different pedo-climatic conditions. Therefore, the differences between ORG and CON are not only due to the effect of management but also to the effect of the site (e.g. soil properties, water balances, etc.).

As a result, several variables from the statistical analyses were not susceptible of comparison between sites, as error mean square were too high to make that comparison in statistical terms. We did not see this lack of comparison between management types as a negative aspect, as our first aim was to determine if old wheat varieties could exert a relevant role in rainfed Mediterranean agroecosystems, both under and organic and under conventional farming conditions.

Point 6: Section 3.2. Rewrite this section, see previous comments.

Response 6: Done.

Point 7: Figure 3. Include the standard error In the bars.

Response 7: Error bars were already included in the original manuscript. However, since they are very small, we have tried to improve the resolution of the graphic.

Point 8: References.  The number of references is extremely high. I suggest authors review the paper trying to reduce the number of references.

Response 8: We have removed 7 references.

The paper made by Guiomar Carranza-Gallego et al. is very relevant focusing on a topic of enormous interest from the agro-ecological point of view. The study is well designed and the parameters analyzed well selected. I have some specific comments related to the landraces and modern varieties used, and some remarks related to the results presented.

Reviewer 2 Report

The originality of the study and the novelty it brings in the field is of actuality. The paper is well structured, the abstract is concise and in the topic; the introduction is supported by well selected bibliographic data. The Experimental and Modeling Approach correctly. Results and Discussions could be improved by studying other papers in the field. English language and style are fine/minor spell check required.

Recommendation -A brief description of varieties taken in the study  by adding information related to sampling, varieties details (origins, size, botany of plant, etc.) and the soil characteristics and termopluviometric parameters as well as a description of agronomic trial.

Author Response

The originality of the study and the novelty it brings in the field is of actuality. The paper is well structured, the abstract is concise and in the topic; the introduction is supported by well selected bibliographic data. The Experimental and Modeling Approach correctly. Results and Discussions could be improved by studying other papers in the field. English language and style are fine/minor spell check required.

Point 1: Recommendation -A brief description of varieties taken in the study by adding information related to sampling, varieties details (origins, size, botany of plant, etc.) and the soil characteristics and termopluviometric parameters as well as a description of agronomic trial.

Response 1: Soil physico-chemical properties of the field trials are shown in Table 1, and data regarding annual rainfall and mean annual temperature during the years of the study are shown in Table 2 in Section 2.1.

We have included a table with the phenological variables of landraces available at CRF web site and from our field experiments in the supplementary data.

Reviewer 3 Report

The document is very interesting because it analyses the importance of landraces (in this case wheat). Knowing the potential of adaptation to climate change is very important for the sustainability of food security in the world. For that reason I see in this work very necessary to be published. By giving importance to local species also gives way to good agricultural practices, not only for increased production, but also for quality to improve product quality.

I recommend to improve figure 1.

Author Response

The document is very interesting because it analyses the importance of landraces (in this case wheat). Knowing the potential of adaptation to climate change is very important for the sustainability of food security in the world. For that reason I see in this work very necessary to be published. By giving importance to local species also gives way to good agricultural practices, not only for increased production, but also for quality to improve product quality.

Point 1: I recommend to improve figure 1.

Response 1: Done. We have tried to improve it through the inclusion of an European map to locate the study area and the border between Portugal and Spain.

Round 2

Reviewer 2 Report

The manuscript have been improved according reviewers suggestions.

Reviewer 3 Report

The authors took into account the suggestions and the article is ready.